# Adaptation to Easy Data in Prediction with Limited Advice

**Tobias Sommer Thune**
Department of Computer Science
University of Copenhagen
tobias.thune@di.ku.dk

**Yevgeny Seldin**
Department of Computer Science
University of Copenhagen
seldin@di.ku.dk

## Abstract

We derive an online learning algorithm with improved regret guarantees for "easy" loss sequences. We consider two types of "easiness": (a) stochastic loss sequences and (b) adversarial loss sequences with small effective range of the losses. While a number of algorithms have been proposed for exploiting small effective range in the full information setting, Gerchinovitz and Lattimore [2016] have shown the impossibility of regret scaling with the effective range of the losses in the bandit setting. We show that just one additional observation per round is sufficient to circumvent the impossibility result. The proposed *Second Order Difference Adjustments* (SODA) algorithm requires no prior knowledge of the effective range of the losses, $\varepsilon$, and achieves an $O(\varepsilon\sqrt{KT\ln K}) + \tilde{O}(\varepsilon K \sqrt[4]{T})$ expected regret guarantee, where $T$ is the time horizon and $K$ is the number of actions. The scaling with the effective loss range is achieved under significantly weaker assumptions than those made by Cesa-Bianchi and Shamir [2018] in an earlier attempt to circumvent the impossibility result. We also provide a regret lower bound of $\Omega(\varepsilon\sqrt{TK})$, which almost matches the upper bound. In addition, we show that in the stochastic setting SODA achieves an $O\left(\sum_{a:\Delta_a>0} \frac{K\varepsilon^2}{\Delta_a}\right)$ pseudo-regret bound that holds simultaneously with the adversarial regret guarantee. In other words, SODA is safe against an unrestricted oblivious adversary and provides improved regret guarantees for at least two different types of "easiness" simultaneously.

## 1 Introduction

Online learning algorithms with both worst-case regret guarantees and refined guarantees for "easy" loss sequences have come into research focus in recent years. In our work we consider *prediction with limited advice* games [Seldin et al., 2014], which are an interpolation between *full information* games [Vovk, 1990, Littlestone and Warmuth, 1994, Cesa-Bianchi and Lugosi, 2006] and games with *limited* (a.k.a. *bandit*) *feedback* [Auer et al., 2002b, Bubeck and Cesa-Bianchi, 2012].[1] In prediction with limited advice the learner faces $K$ unobserved sequences of losses $\{\ell_t^a\}_{t,a}$, where $a$ indexes the sequence number and $t$ indexes the elements within the $a$-th sequence. At each round $t$ of the game the learner picks a sequence $A_t \in \{1, \ldots, K\}$ and suffers the loss $\ell_t^{A_t}$, which is then observed. After that, the learner is allowed to observe the losses of $M$ additional sequences in the same round $t$, where $0 \leq M \leq K-1$. For $M = K-1$ the setting is equivalent to a full information game and for $M = 0$ it becomes a bandit game.

For a practical motivation behind prediction with limited advice imagine that the loss sequences correspond to losses of $K$ different algorithms for solving some problem, or $K$ different parametrizations of one algorithm, or $K$ different experts. If we had the opportunity we would have executed all the algorithms or queried all the experts before making a prediction. This would correspond to a full information game. But in reality we may be constrained by time, computational power, or monetary budget. In such case we are forced to select algorithms or experts to query. Being able to query just one expert or algorithm per prediction round corresponds to a bandit game, but we may have time or money to get a bit more, even though not all of it. This is the setting modeled by prediction with limited advice.

Our goal is to derive an algorithm for prediction with limited advice that is robust in the worst case and provides improved regret guarantees in "easy" cases. There are multiple ways to define "easiness" of loss sequences. Among them, loss sequences generated by i.i.d. sources, like the classical stochastic bandit model [Robbins, 1952, Lai and Robbins, 1985, Auer et al., 2002a], and adversarial sequences with bounded effective range of the losses within each round [Cesa-Bianchi et al., 2007]. For the former a simple calculation shows that in the full information setting the basic Hedge algorithm [Vovk, 1990, Littlestone and Warmuth, 1994] achieves an improved "constant" (independent of time horizon) pseudo-regret guarantee without sacrificing the worst-case guarantee. Much more work is required to achieve adaptation to this form of easiness in the bandit setting if we want to keep the adversarial regret guarantee simultaneously [Bubeck and Slivkins, 2012, Seldin and Slivkins, 2014, Auer and Chiang, 2016, Seldin and Lugosi, 2017, Wei and Luo, 2018, Zimmert and Seldin, 2018].

An algorithm that adapts to the second form of easiness in the full information setting was first proposed by Cesa-Bianchi et al. [2007] and a number of variations have followed [Gaillard et al., 2014, Koolen and van Erven, 2015, Luo and Schapire, 2015, Wintenberger, 2017]. However, a recent result by Gerchinovitz and Lattimore [2016] have shown that such adaptation is impossible in the bandit setting. Cesa-Bianchi and Shamir [2018] proposed a way to circumvent the impossibility result by either assuming that the ranges of the individual losses are provided to the algorithm in advance or assuming that the losses are smooth and an "anchor" loss of one additional arm is provided to the algorithm. The latter assumption has so far only lead to a substantial improvement when the "anchor" loss is always the smallest loss in the corresponding round.

We consider adaptation to both types of easiness in prediction with limited advice. We show that $M = 1$ (just one additional observation per round) is sufficient to circumvent the impossibility result of Gerchinovitz and Lattimore [2016]. This assumption is weaker than the assumptions in Cesa-Bianchi and Shamir [2018]. We propose an algorithm, which achieves improved regret guarantees both when the effective loss range is small and when the losses are stochastic (generated i.i.d.). The algorithm is inspired by the BOA algorithm of Wintenberger [2017], but instead of working with exponential weights of the cumulative losses and their second moment corrections it uses estimates of the loss differences. The algorithm achieves an $O(\varepsilon\sqrt{KT \ln K}) + \tilde{O}(\varepsilon K \sqrt[4]{T})$ expected regret guarantee with no prior knowledge of the effective loss range $\varepsilon$ or time horizon $T$. We also provide regret lower bound of $\Omega(\varepsilon\sqrt{KT})$, which matches the upper bound up to logarithmic terms and smaller order factors. Furthermore, we show that in the stochastic setting the algorithm achieves an $O\left(\sum_{a:\Delta_a > 0} \frac{K\varepsilon^2}{\Delta_a}\right)$ pseudo-regret guarantee. The improvement in the stochastic setting is achieved without compromising the adversarial regret guarantee.

The paper is structured in the following way. In Section 2 we lay out the problem setting. In Section 3 we present the algorithm and in Section 4 the main results about the algorithm. Proofs of the main results are presented in Section 5.

## 2  Problem Setting

We consider sequential games defined by $K$ infinite sequences of losses $\{\ell_1^a, \ell_2^a, \dots\}_{a \in \{1,\dots,K\}}$, where $\ell_t^a \in [0, 1]$ for all $a$ and $t$. At each round $t \in \{1, 2, \dots\}$ of the game the learner selects an action (a.k.a. "arm") $A_t \in [K] := \{1, \dots, K\}$ and then suffers and observes the corresponding loss $\ell_t^{A_t}$. Additionally, the learner is allowed to choose a second arm, $B_t$, and observe $\ell_t^{B_t}$. The loss of the second arm, $\ell_t^{B_t}$, is not suffered by the learner. (This is analogous to the full information setting, where the losses of all arms $a \neq A_t$ are observed, but not suffered). It is assumed that $\ell_t^{B_t}$ is observed

*after* $A_t$ has been selected, but other relative timing of events within a round is unimportant for our analysis.

The performance of the learner up to round $T$ is measured by *expected regret* defined as

$$\mathcal{R}_T := \mathbb{E}\left[\sum_{t=1}^{T} \ell_t^{A_t}\right] - \min_{a \in [K]} \mathbb{E}\left[\sum_{t=1}^{T} \ell_t^a\right], \tag{1}$$

where the expectation is taken with respect to potential randomization of the loss generation process and potential randomization of the algorithm. We note that in the adversarial setting the losses are considered deterministic and the second expectation can be omitted, whereas in the stochastic setting the definition coincides with the definition of pseudo-regret [Bubeck and Cesa-Bianchi, 2012, Seldin and Lugosi, 2017]. In some literature $\mathcal{R}_T$ is termed *excess of cumulative predictive risk* [Wintenberger, 2017].

Below we define adversarial and stochastic loss generation models and effective range of loss sequences.

### Adversarial losses

In the adversarial setting the loss sequences are selected arbitrarily by an adversary. We restrict ourselves to the *oblivious* model, where the losses are fixed before the start of the game and do not depend on the actions of the learner.

### Stochastic losses

In the stochastic setting the losses are drawn i.i.d., so that $\mathbb{E}[\ell_t^a] = \mu_a$ independently of $t$. Since we have a finite number of arms, there exists a best arm $a^\star$ (not necessarily unique) such that $\mu_{a^\star} \leq \mu_a$ for all $a$. We further define the suboptimality *gaps* by

$$\Delta_a := \mu_a - \mu_{a^\star} \geq 0.$$

In the stochastic setting the expected regret can be rewritten as

$$\mathcal{R}_T = \sum_{a \in [K]: \Delta_a > 0} \Delta_a \, \mathbb{E}\left[\sum_{t=1}^{T} \mathbb{1}(A_t = a)\right], \tag{2}$$

where $\mathbb{1}$ is the indicator function.

### Effective loss range

For both the adversarial and stochastic losses, we define the *effective loss range* as the smallest number $\varepsilon$, such that for all $t \in [T]$ and $a, a' \in [K]$:

$$|\ell_t^a - \ell_t^{a'}| \leq \varepsilon \quad \text{almost surely.} \tag{3}$$

Since we have assumed that $\ell_t^a \in [0, 1]$, we have $\varepsilon \leq 1$, where $\varepsilon = 1$ corresponds to an unrestricted setting.

## 3 Algorithm

We introduce the *Second Order Difference Adjustments* (*SODA*) algorithm, summarized in Algorithm 1. SODA belongs to the general class of *exponential weights* algorithms. The algorithm has two important distinctions from the common members of this class. First, it uses cumulative *loss difference estimators* instead of cumulative loss estimators for the exponential weights updates. Instantaneous loss difference estimators at round $t$ are defined by

$$\widetilde{\Delta \ell}_t^a = (K-1)\mathbb{1}(B_t = a)\left(\ell_t^{B_t} - \ell_t^{A_t}\right). \tag{4}$$

SODA samples the "secondary" action $B_t$ (the additional observation) uniformly from $K-1$ arms, all except $A_t$, and the $(K-1)$ term above corresponds to importance weighting with respect to the

sampling of $B_t$. The loss difference estimators scale with the effective range of the losses and they can be positive and negative. Both of these properties are distinct from the traditional loss estimators. The second difference is that we are using a second order adjustment in the weighting inspired by Wintenberger [2017]. We define the cumulative loss difference estimator and its second moment by

$$D_t(a) := \sum_{s=1}^{t} \widetilde{\Delta\ell}_s^a, \quad S_t(a) := \sum_{s=1}^{t} \left( \widetilde{\Delta\ell}_s^a \right)^2. \tag{5}$$

We then have the distribution $\boldsymbol{p_t}$ for selecting the primary action $A_t$ defined by

$$p_t^a = \frac{\exp\left(-\eta_t D_{t-1}(a) - \eta_t^2 S_{t-1}(a)\right)}{\sum_{a=1}^{K} \exp\left(-\eta_t D_{t-1}(a) - \eta_t^2 S_{t-1}(a)\right)}, \tag{6}$$

where $\eta_t$ is a learning rate scheme, defined as

$$\eta_t = \min \left\{ \sqrt{\frac{\ln K}{\max_a S_{t-1}(a) + (K-1)^2}}, \frac{1}{2(K-1)} \right\}. \tag{7}$$

The learning rate satisfies $\eta_t \le 1/(2\varepsilon(K-1))$ for all $t$, which is required for the subsequent analysis.

The algorithm is summarized below:

---

Initialize $\boldsymbol{p_1} \leftarrow (1/K, \dots, 1/K)$.
**for** $t = 1, 2, \dots$ **do**
  Draw $A_t$ according to $\boldsymbol{p_t}$;
  Draw $B_t$ uniformly at random from the remaining actions $[K] \setminus \{A_t\}$;
  Observe $\ell_t^{A_t}, \ell_t^{B_t}$ and suffer $\ell_t^{A_t}$;
  Construct $\widetilde{\Delta\ell}_t^a$ by equation (4);
  Update $D_t(a), S_t(a)$ by (5);
  Define $\boldsymbol{p_{t+1}}$ by (6);
**end**

**Algorithm 1:** Second Order Difference Adjustments (SODA)

---

## 4 Main Results

We are now ready to present the regret bounds for SODA. We start with regret upper and lower bounds in the adversarial regime and then show that the algorithm simultaneously achieves improved regret guarantee in the stochastic regime.

### 4.1 Regret Upper Bound in the Adversarial Regime

First we provide an upper bound for the expected regret of SODA against oblivious adversaries that produce loss sequences with effective loss range bounded by $\varepsilon$. Note that this result does not depend on prior knowledge of the effective loss range $\varepsilon$ or time horizon $T$.

**Theorem 1.** *The expected regret of* SODA *against an oblivious adversary satisfies*

$$\mathcal{R}_T \le 4\varepsilon\sqrt{(K-1)\ln K} \sqrt{T + (K-1)\sqrt{T}\left(2 + \sqrt{\ln\left(\sqrt{T}(K-1)\right)/2}\right)} + 4(K-1)\ln K.$$

A proof of this theorem is provided in Section 5.1.[2] The upper bound scales as $O(\varepsilon\sqrt{KT\ln K}) + \tilde{O}(\varepsilon K \sqrt[4]{T})$, which nearly matches the lower bound provided below.

## 4.2 Regret Lower Bound in the Adversarial Regime

We show that in the worst case the regret must scale linearly with the effective loss range $\varepsilon$.

**Theorem 2.** *In prediction with limited advice with $M = 1$ (one additional observation per round or, equivalently, two observations per round in total), for loss sequences with effective loss range $\varepsilon$, we have for $T \geq 3K/32$:*

$$\inf \sup \mathcal{R}_T \geq 0.02\varepsilon\sqrt{KT},$$

*where the infimum is with respect to the choices of the algorithm and the supremum is over all oblivious loss sequences with effective loss range bounded by $\varepsilon$.*

The theorem is proven by adaptation of the $\Omega(\sqrt{KT})$ lower bound by Seldin et al. [2014] for prediction with limited advice with unrestricted losses in $[0, 1]$ and one extra observation. We provide it in Appendix A. Note that the upper bound in Theorem 1 matches the lower bound up to logarithmic terms and lower order additive factors. In particular, changing the selection strategy for the second arm, $B_t$, from uniform to anything more sophisticated is not expected to yield significant benefits in the adversarial regime.

## 4.3 Regret Upper Bound in the Stochastic Regime

Finally, we show that SODA enjoys constant expected regret in the stochastic regime. This is achieved without sacrificing the adversarial regret guarantee.

**Theorem 3.** *The expected regret of* SODA *applied to stochastic loss sequences with gaps $\Delta_a$ satisfies*

$$\mathcal{R}_T \leq \sum_{a:\Delta_a>0} \left[ \left( \frac{8K}{\ln K} + 16 \right) \frac{\varepsilon^2}{\Delta_a} + 4K + \frac{\Delta_a}{K} \right]. \tag{8}$$

A brief sketch of a proof of this theorem is given in Section 5.2, with the complete proof provided in Appendix C.

Note that $\varepsilon$ is the effective range of realizations of the losses, whereas the gaps $\Delta_a$ are based on the expected losses. Naturally, $\Delta_a \leq \varepsilon$. For example, if the losses are Bernoulli then the range is $\varepsilon = 1$, but the gaps are based on the distances between the biases of the Bernoulli variables. When the losses are not $\{0, 1\}$, but confined to a smaller range $\varepsilon$, Theorem 3 yields a tighter regret bound. The scaling of the regret bound in $K$ is suboptimal and it is currently unknown whether it could be improved without compromising the worst-case guarantee. Perhaps changing the selection strategy for $B_t$ could help here. We leave this improvement for future work.

To summarize, SODA achieves adversarial regret guarantee that scales with the effective loss range and almost matches the lower bound and simultaneously has improved regret guarantee in the stochastic regime.

# 5 Proofs

This section contains the proof of Theorem 1 and a proof sketch for Theorem 3. The proof of Theorem 2 is provided in Appendix A.

## 5.1 Proof of Theorem 1

The proof of the theorem is prefaced by two lemmas, but first we show some properties of the loss difference estimators. We use $\mathbb{E}_{B_t}$ to denote expectation with respect to selection of $B_t$ conditioned on all random outcomes prior to this selection. For oblivious adversaries, the expected cumulative loss difference estimators are equal to the negative expect regret against the corresponding arm $a$:

$$\mathbb{E}\left[\sum_{t=1}^{T} \widetilde{\Delta\ell}_t^a\right] = \mathbb{E}\left[\sum_{t=1}^{T} \mathbb{E}_{B_t}\left[\widetilde{\Delta\ell}_t^a\right]\right] = \mathbb{E}\left[\sum_{t=1}^{T}\left(\ell_t^a - \ell_t^{A_t}\right)\right] = \sum_{t=1}^{T}\ell_t^a - \mathbb{E}\left[\sum_{t=1}^{T}\ell_t^{A_t}\right] =: -\mathcal{R}_T^a,$$

where we have used the fact that $\widetilde{\Delta \ell}_t^a$ is an unbiased estimate of $\ell_t^a - \ell_t^{A_t}$ due to importance weighting with respect to the choice of $B_t$. Similarly, we have

$$\mathbb{E}\left[\sum_{t=1}^{T}\left(\widetilde{\Delta \ell}_t^a\right)^2\right] = (K-1)\ \mathbb{E}\left[\sum_{t=1}^{T}\left(\ell_t^a - \ell_t^{A_t}\right)^2\right]. \tag{9}$$

Similar to the analysis of the anytime version of EXP3 in Bubeck and Cesa-Bianchi [2012], which builds on Auer et al. [2002b], we consider upper and lower bounds on the expectation of the incremental update. This is captured by the following lemma:

**Lemma 1.** *With a learning rate scheme $\eta_t$ for $t = 1, 2, \ldots$, where $\eta_t \leq 1/2\varepsilon(K-1)$, SODA fulfills:*

$$-\sum_{t=1}^{T} \widetilde{\Delta \ell}_t^a \leq \frac{\ln K}{\eta_T} + \eta_T \sum_{t=1}^{T}\left(\widetilde{\Delta \ell}_t^a\right)^2 - \sum_{t=1}^{T} \underset{a \sim p_t}{\mathbb{E}}\left[\widetilde{\Delta \ell}_t^a\right] + \sum_{t}\left(\Phi_t(\eta_{t+1}) - \Phi_t(\eta_t)\right) \tag{10}$$

*for all a, where we define the* potential

$$\Phi_t(\eta) := \frac{1}{\eta}\ln\left(\frac{1}{K}\sum_{a=1}^{K}\exp\left(-\eta D_t(a) - \eta^2 S_t(a)\right)\right). \tag{11}$$

Note that unlike in the analysis of EXP3, here the learning rates $\eta_t$ do not have to be non-increasing. A proof of this lemma is based on modification of standard arguments and is found in Appendix B.1.

The second lemma is a technical one and is proven in Appendix B.2.

**Lemma 2.** *Let $\sigma_t$ with $t \in \mathbb{N}$ be an increasing positive sequence with bounded differences such that $\sigma_t - \sigma_{t-1} \leq c$ for a finite constant c. Let further $\sigma_0 = 0$. Then*

$$\sum_{t=1}^{T} \sigma_t\left(\frac{1}{\sqrt{\sigma_{t-1} + c}} - \frac{1}{\sqrt{\sigma_t + c}}\right) \leq 2\sqrt{\sigma_{T-1} + c}.$$

**Proof of Theorem 1** We apply Lemma 1, which leads to the following inequality for any learning rate scheme $\eta_t$ for $t = 1, 2, \ldots$, where $\eta_t \leq 1/2\varepsilon(K-1)$:

$$-\sum_{t=1}^{T} \widetilde{\Delta \ell}_t^a \leq \underbrace{\frac{\ln K}{\eta_T}}_{1^{\text{st}}} + \underbrace{\eta_T \sum_{t=1}^{T}\left(\widetilde{\Delta \ell}_t^a\right)^2}_{2^{\text{nd}}} - \underbrace{\sum_{t=1}^{T} \underset{a \sim p_t}{\mathbb{E}}\left[\widetilde{\Delta \ell}_t^a\right]}_{3^{\text{rd}}} + \underbrace{\sum_{t=1}^{T}\left(\Phi_t(\eta_{t+1}) - \Phi_t(\eta_t)\right)}_{4^{\text{th}}}. \tag{12}$$

Note that in expectation, the left hand side of (12) is the regret against arm $a$. We are thus interested in bounding the expectation of the terms on the right hand side, where we note that the third term vanishes in expectation. We first consider the case where $\eta_t = \sqrt{\ln K/(\max_a S_t(a) + (K-1)^2)}$, postponing the initial value for now.

The first term becomes:

$$\frac{\ln K}{\eta_T} = \sqrt{\ln K}\sqrt{\max_a S_{T-1}(a) + (K-1)^2}. \tag{13}$$

The second term becomes:

$$\eta_T S_T(a) = \sqrt{\ln K}\frac{S_T(a)}{\sqrt{\max_a S_{T-1}(a) + (K-1)^2}} \leq \sqrt{\ln K}\sqrt{\max_a S_{T-1}(a) + (K-1)^2}, \tag{14}$$

where we use that $S_t(a) \leq S_{t-1}(a) + (K-1)^2$ for all $t$ by design.

Finally, for the fourth term in equation (12), we need to consider the potential differences. Unlike in the anytime analysis of EXP3, where this term is negative [Bubeck and Cesa-Bianchi, 2012], in our case it turns to be related to the second moment of the loss difference estimators. We let

$$q_t^\eta = \frac{\exp\left(-\eta D_t(a) - \eta^2 S_t(a)\right)}{\sum_{a=1}^{K}\exp\left(-\eta D_t(a) - \eta^2 S_t(a)\right)} \tag{15}$$

denote the exponential update using the loss estimators up to $t$, but with a free learning rate $\eta$. We further suppress some indices for readability, such that $D_a = D_t(a)$ and $S_a = S_t(a)$ in the following. We have

$$\Phi'_t(\eta) = -\frac{1}{\eta^2}\ln\left(\frac{1}{K}\sum_a \exp\left(-\eta D_a - \eta^2 S_a\right)\right) + \frac{1}{\eta}\frac{\sum_a \exp\left(-\eta D_a - \eta^2 S_a\right)\cdot\left(-D_a - 2\eta S_a\right)}{\sum_a \exp\left(-\eta D_a - \eta^2 S_a\right)}$$

$$= \frac{\sum_a\left(\exp\left(-\eta D_a - \eta^2 S_a\right)\cdot\left(-\eta D_a - 2\eta^2 S_a - \ln\left(\frac{1}{K}\sum_a \exp\left(-\eta D_a - \eta^2 S_a\right)\right)\right)\right)}{\eta^2\sum_a \exp\left(-\eta D_a - \eta^2 S_a\right)}.$$

By using $-\eta D_a - 2\eta^2 S_a = \ln\left(\exp(-\eta D_a - \eta^2 S_a)\exp(-\eta^2 S_a)\right)$ the above becomes

$$\Phi'_t(\eta) = \frac{1}{\eta^2}\mathop{\mathbb{E}}_{a\sim q_t^\eta}\left[\ln\left(\frac{q_t^\eta(a)}{1/K}\exp(-\eta^2 S_a)\right)\right] = \frac{1}{\eta^2}\mathrm{KL}\left(q_t^\eta\|\mathbf{1}/\boldsymbol{K}\right) - \mathop{\mathbb{E}}_{a\sim q_t^\eta}\left[S_t(a)\right], \qquad (16)$$

where we have used that $\mathbf{1}/\boldsymbol{K}$ is the pmf. of the uniform distribution over $K$ arms. Since the KL-divergence is always positive, we can rewrite the potential differences as

$$\Phi_t(\eta_{t+1}) - \Phi_t(\eta_t) = -\int_{\eta_{t+1}}^{\eta_t}\Phi'_t(\eta)d\eta \leq \int_{\eta_{t+1}}^{\eta_t}\mathop{\mathbb{E}}_{a\sim q_t^\eta}\left[S_t(a)\right]d\eta \leq \int_{\eta_{t+1}}^{\eta_t}\max_a S_t(a)d\eta$$

$$= \sqrt{\ln K}\max_a S_t(a)\left(\frac{1}{\sqrt{\max\limits_a S_{t-1}(a) + (K-1)^2}} - \frac{1}{\sqrt{\max\limits_a S_t(a) + (K-1)^2}}\right).$$

By Lemma 2 we then have

$$\sum_{t=1}^{T}\Phi_t(\eta_{t+1}) - \Phi_t(\eta_t) \leq 2\sqrt{\ln K}\sqrt{\max_a S_{T-1}(a) + (K-1)^2}. \qquad (17)$$

Collecting the terms (13), (14) and (17) and noting that these bounds hold for all $a$, by taking expectations and using Jensen's inequality we get

$$\mathcal{R}_T \leq \mathbb{E}\left[4\sqrt{\ln K}\sqrt{\max_a S_{T-1}(a) + (K-1)^2}\right]$$

$$\leq 4\sqrt{\ln K}\sqrt{\mathbb{E}\left[\max_a S_{T-1}(a)\right] + (K-1)^2}. \qquad (18)$$

The remainder of the proof is to bound this inner expectation:

$$\mathbb{E}\left[\max_a S_{T-1}(a)\right] \leq (K-1)^2\varepsilon^2\,\mathbb{E}\left[\max_a\sum_{t=1}^{T-1}\mathbb{1}[B_t = a]\right].$$

Let $Z_t^a = \sum_{s=1}^{t}\mathbb{1}[B_s = a]$ and note that $Z_{T-1}^a \leq T-1$. We now consider a partioning of the probability for a cutoff $\alpha > 0$:

$$\mathbb{E}[\max_a Z_{T-1}^a] \leq \alpha\,\mathbb{P}\left\{\max_a Z_{T-1}^a \leq \alpha\right\} + (T-1)\,\mathbb{P}\left\{\max_a Z_{T-1}^a > \alpha\right\}$$

$$\leq \alpha + (T-1)K\,\mathbb{P}\left\{Z_{T-1}^a > \alpha\right\},$$

using a union bound for the final inequality. To continue we need to address the fact that the $B_t$'s are not independent. We can however note that $\mathbb{P}\{B_t = a\} \leq (K-1)^{-1}$ for all $t$ and $a$. By letting $x_t^a$ be Bernoulli with parameter $(K-1)^{-1}$ and $X_T^a = \sum_{t=1}^{T}x_t^a$ we then get

$$\mathbb{P}\left\{Z_{T-1}^a > \alpha\right\} \leq \mathbb{P}\left\{X_{T-1}^a > \alpha\right\}. \qquad (19)$$

In the upper bound we can thus substitute $X_{T-1}^a$ for $Z_{T-1}^a$ and exploit the fact that the $x_t^a$'s are independent by construction. Note further that $\mathbb{E}[X_{T-1}^a] = \frac{T-1}{K-1}$, so by choosing $\alpha = \frac{T-1}{K-1} + \delta$ for

$\delta > 0$, we obtain by Hoeffding's inequality:

$$\mathbb{E}[\max_a Z_{T-1}^a] \leq \frac{T-1}{K-1} + \delta + (T-1)K\,\mathbb{P}\left\{X_{T-1}^a - \frac{T-1}{K-1} > \delta\right\}$$

$$\leq \frac{T-1}{K-1} + \delta + (T-1)K\exp\left(-\frac{2\delta^2}{T-1}\right).$$

We now choose $\delta = \sqrt{\frac{T}{2}\ln\left(\sqrt{T}(K-1)\right)}$, which gives us

$$\mathbb{E}[\max_a Z_{T-1}^a] \leq \frac{T-1}{K-1} + \sqrt{\frac{T}{2}\ln\left(\sqrt{T}(K-1)\right)} + 2\sqrt{T}.$$

Inserting this in (18) gives us the desired bound.

For the case where the learning rate at $T$ is instead given by $1/2(K-1)$ implying $4(K-1)^2\ln K \geq \max_a S_{T-1}(a) + (K-1)^2$, the first term is $\frac{\ln K}{\eta_T} = 2(K-1)\ln K$, and the second term is

$$\eta_T S_T(a) = \frac{1}{2(K-1)} S_T(a) \leq \frac{S_{T-1}(a) + (K-1)^2}{2(K-1)} \leq \frac{4(K-1)^2\ln K}{2(K-1)} \leq 2(K-1)\ln K.$$

Since the learning rate is constant the potential differences vanish, completing the proof. $\square$

## 5.2 Proof sketch of Theorem 3

Here we present the key ideas used to prove Theorem 3. The complete proof is provided in Appendix C.

Recall that the expected regret in the stochastic setting is given by (2), where $\mathbb{E}[\mathbb{1}(A_t = a)] = \mathbb{E}[p_t^a]$. Thus, we need to bound $\mathbb{E}[\sum_t p_t^a]$. The first step is to bound this as

$$\mathbb{E}[p_t^a] \leq \sigma + \mathbb{P}\{p_t^a > \sigma\} \leq \sigma + \mathbb{P}\left\{Ke^{-\eta_t\sum_{i=1}^{t-1}X_i} > \sigma\right\} \qquad (20)$$

for a positive threshold $\sigma$, where we show that $p_t^a \leq Ke^{-\eta_t\sum_{i=1}^{t-1}X_i}$ for $X_i := \widetilde{\Delta\ell}_i^a - \widetilde{\Delta\ell}_i^{a^\star}$. This approach is motivated by the fact that $\mathbb{E}_{B_i}[\widetilde{\Delta\ell}_i^a - \widetilde{\Delta\ell}_i^{a^\star}] \propto \Delta_a$, where the expectation is with respect to selection of $B_i$ and the loss generation, conditioned on all prior randomness.

The next step is to tune $\sigma \propto \exp(\sum\mathbb{E}_i[X_i])$, which allows us to bound the second term using Azuma's inequality and balance the two terms. Finally, this bound is summed over $t$ using a technical lemma for the limit of this sum.

# 6 Discussion

We have presented the SODA algorithm for prediction with limited advice with two observations per round (the "primary" observation of the loss of the action that was played and one additional observation). We have shown that the algorithm adapts to two types of simplicity of loss sequences simultaneously: (a) it provides improved regret guarantees for adversarial sequences with bounded effective range of the losses and (b) for stochastic loss sequences. In both cases the regret scales linearly with the effective range and the knowledge of the range is not required. In the adversarial case we achieve $O(\varepsilon\sqrt{KT\ln K}) + \tilde{O}(\varepsilon K \sqrt[4]{T})$ regret guarantee and in the stochastic case we achieve $O\left(\sum_{a:\Delta_a > 0}\frac{K\varepsilon^2}{\Delta_a}\right)$ regret guarantee. Our result demonstrates that just one extra observation per round is sufficient to circumvent the impossibility result of Gerchinovitz and Lattimore [2016] and significantly relaxes the assumptions made by Cesa-Bianchi and Shamir [2018] to achieve the same goal.

There are a number of open questions and interesting directions for future research. One is to improve the regret guarantee in the stochastic regime. Another is to extend the results to bandits with limited advice in the spirit of Seldin et al. [2013], Kale [2014].

**Acknowledgments**

The authors thank Julian Zimmert for valuable input and discussions.

## Footnotes

[1]There exists an orthogonal interpolation between full information and bandit games through the use of feedback graphs Alon et al. [2017], which is different and incomparable with prediction with limited advice, see Seldin et al. [2014] for a discussion.

[2]It is straightforward to extended the analysis to time-varying ranges, $\varepsilon_t : |\ell_t^a - \ell_t^{a'}| \le \varepsilon_t$ for all $a, a'$ a.s., which leads to an $O\left(\sqrt{\sum_{t=1}^{T}(\varepsilon_t^2)K\ln K}\right) + \tilde{O}\left(K\sqrt[4]{\sum_{t=1}^{T}\varepsilon_t^2}\right)$ regret bound . For the sake of clarity we restrict the presentation to a constant $\varepsilon$.

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
