[Supplementary Material · Thune_Seldin_Supplementary.pdf]

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

# A  Proof of Theorem 2

The lower bound is a straightforward adaptation of Theorem 2 in Seldin et al. [2014], which states that for prediction with limited advice where $M' = M + 1$ of $K$ experts are queried, we have for $T \geq \frac{3}{16}\frac{K}{M'}$:

$$\inf \sup \mathcal{R}_T \geq 0.03\sqrt{\frac{K}{M'T}},$$

where the infimum is over learning strategies and the supremum over oblivious adversaries.

Our case of $M = 1$ additional expert corresponds to $M' = 2$. The proof of the above is based upon the standard technique for lower bounding, where Bernoulli losses with varying biases are constructed. As this is a stochastic setting, the regret of playing a suboptimal arm $a$ is analysed as

$$(\nu_a - \nu_{a^\star})\,\mathbb{E}[N_T(a)],$$

where the $\nu$'s are the biases of the Bernoulli variables and $N_T(a)$ is the number of times an arm is played. The rest of analysis consists of lower bounding the expected number of plays and tuning the biases.

By changing the constructed losses to Bernoulli variables times $\varepsilon$ (i.e. taking values in $\{0, \varepsilon\}$), the expected values become $\varepsilon\nu_a$, which means we get a factor of $\varepsilon$ in the above expression. Since the bound on $\mathbb{E}[N_T(a)]$ does not depend on the values taken by the distributions, but only the ability to discern them, the proof follows directly from that in Seldin et al. [2014]. $\qquad\square$

# B  Supplement for the proof of Theorem 1 (Section 5.1)

## B.1  Proof of Lemma 1

We first derive two inequalities, which are combined and rearranged into the statement of the lemma.

Consider the quantity

$$\sum_{t=1}^{T}\frac{1}{\eta_t}\ln\mathop{\mathbb{E}}_{a\sim p_t}\left[\exp\left(-\eta_t\widetilde{\Delta\ell}_t^a - \eta_t^2\left(\widetilde{\Delta\ell}_t^a\right)^2\right)\right] \leq \sum_{t=1}^{T}\frac{1}{\eta_t}\ln\mathop{\mathbb{E}}_{a\sim p_t}\left[1 - \eta_t\widetilde{\Delta\ell}_t^a\right]$$

$$= \sum_{t=1}^{T}\frac{1}{\eta_t}\ln\left(1 - \eta_t\mathop{\mathbb{E}}_{a\sim p_t}\left[\widetilde{\Delta\ell}_t^a\right]\right)$$

$$\leq -\sum_{t=1}^{T}\mathop{\mathbb{E}}_{a\sim p_t}\left[\widetilde{\Delta\ell}_t^a\right],$$

where the first step is based on the inequality $e^{z-z^2} \leq 1 + z$ for $z = -\eta_t\widetilde{\Delta\ell}_t^a \geq -1/2$ [Cesa-Bianchi et al., 2007]. The upper bound on $\eta_t \leq (2\varepsilon(K-1))^{-1}$ guarantees that the condition of the inequality holds. The last step is based on $\ln(1+z) \leq z$ for $z > -1$.

Using the potential (11) we can rewrite the same quantity as

$$\frac{1}{\eta_t}\ln\mathop{\mathbb{E}}_{a\sim p_t}\left[\exp\left(-\eta_t\widetilde{\Delta\ell}_t^a - \eta_t^2\left(\widetilde{\Delta\ell}_t^a\right)^2\right)\right] = \frac{1}{\eta_t}\ln\sum_{a=1}^{K}\exp\left(-\eta_t\widetilde{\Delta\ell}_t^a - \eta_t^2\left(\widetilde{\Delta\ell}_t^a\right)^2\right)\cdot p_t^a$$

$$= \frac{1}{\eta_t}\ln\frac{\sum_{a=1}^{K}\exp\left(-\eta_t D_t(a) - \eta_t^2 S_t(a)\right)}{\sum_{a=1}^{K}\exp\left(-\eta_t D_{t-1}(a) - \eta_t^2 S_{t-1}(a)\right)}$$

$$= \Phi_t(\eta_t) - \Phi_{t-1}(\eta_t).$$

Summing over $t$ and reindexing the sum we get

$$\sum_{t=1}^{T}\left(\Phi_t(\eta_t) - \Phi_{t-1}(\eta_t)\right) = \sum_{t=1}^{T-1}\left(\Phi_t(\eta_t) - \Phi_t(\eta_{t+1})\right) + \Phi_T(\eta_T) - \Phi_0(\eta_1).$$

Since by definition $D_0 = 0$ and $S_0 = 0$, we have $\Phi_0(\eta_1) = 0$. Next, we lower bound the middle term:

$$\Phi_T(\eta_T) = \frac{1}{\eta_T} \ln \left( \frac{1}{K} \sum_{a=1}^K \exp\left(-\eta_T D_T(a) - \eta_T^2 S_T(a)\right) \right)$$

$$\geq -\frac{\ln K}{\eta_T} + \frac{1}{\eta_T} \ln \left( \exp\left(-\eta_T D_T(a) - \eta_T^2 S_T(a)\right)\right)$$

$$= -\frac{\ln K}{\eta_T} - D_T(a) - \eta_T S_T(a),$$

where we have used that the logarithm is monotonously increasing and all the terms in the inner sum are positive.

By using the lower and upper bounds simultaneously and moving everything except for $-D_T(a)$ from the left hand side, the proof is complete. □

### B.2 Proof of Lemma 2

By the boundedness we have:

$$\sum_{t=1}^T \sigma_t \left( \frac{1}{\sqrt{\sigma_{t-1} + c}} - \frac{1}{\sqrt{\sigma_t + c}} \right) \leq \sum_{t=1}^T (\sigma_{t-1} + c) \left( \frac{1}{\sqrt{\sigma_{t-1} + c}} - \frac{1}{\sqrt{\sigma_t + c}} \right)$$

$$= \sum_{t=0}^{T-1} \frac{\sigma_t + c}{\sqrt{\sigma_t + c}} - \sum_{t=1}^T \frac{\sigma_{t-1} + c}{\sqrt{\sigma_t + c}}$$

$$= \sum_{t=1}^{T-1} \frac{\sigma_t - \sigma_{t-1}}{\sqrt{\sigma_t + c}} + \frac{\sigma_0 + c}{\sqrt{\sigma_0 + c}} - \frac{\sigma_{T-1} + c}{\sqrt{\sigma_T + c}}.$$

Here the second term is $\sqrt{c}$ and the third is negative and can thus be discarded in the upper bound. The first term is a lower Riemann sum of $x \mapsto 1/\sqrt{x + c}$, giving us:

$$\sum_{t=1}^T \sigma_t \left( \frac{1}{\sqrt{\sigma_{t-1} + c}} - \frac{1}{\sqrt{\sigma_t + c}} \right) \leq \sqrt{c} + \int_{\sigma_1}^{\sigma_{T-1}} \frac{1}{\sqrt{x + c}} \, dx$$

$$= \sqrt{c} + 2\sqrt{x + c} \Big|_{\sigma_1}^{\sigma_{T-1}}$$

$$\leq 2\sqrt{\sigma_{T-1} + c},$$

where the final inequality uses $2\sqrt{\sigma_1 + c} > \sqrt{c}$. □

## C   Proof of Theorem 3

Before proving the theorem we need the following technical lemma:

**Lemma 3.** *For $c > 0$ we have*

$$\sum_{t=1}^\infty e^{-c\sqrt{t}} \leq \frac{2}{c^2}, \quad \text{and} \quad \sum_{t=1}^\infty e^{-ct} \leq \frac{1}{c}.$$

*Proof.* For the first part, note that

$$\int e^{-c\sqrt{t}} dt = -\frac{2}{c}\sqrt{t}e^{-c\sqrt{t}} - \frac{2}{c^2}e^{-c\sqrt{t}},$$

which is confirmed by differentiation. Then

$$\sum_{t=1}^\infty e^{-c\sqrt{t}} \leq \int_0^\infty e^{-c\sqrt{t}} dt = -\frac{2}{c}\sqrt{t}e^{-c\sqrt{t}} - \frac{2}{c^2}e^{-c\sqrt{t}} \Big|_0^\infty = \frac{2}{c^2},$$

where we use that the summand is decreasing, making the series a lower Riemann sum of the intergral. For the second part we use the exact limit and that $e^x - 1 \geq x$ with the same sign for all $x$:

$$\sum_{t=1}^{\infty} e^{-ct} = \frac{1}{e^c - 1} \leq \frac{1}{c}. \qquad \square$$

**Proof of Theorem 3**  Recall that the expected regret in the stochastic setting is given by

$$\mathcal{R}_T = \sum_{a:\Delta_a > 0} \Delta_a \, \mathbb{E} \left[ \sum_{t=1}^{T} \mathbb{1}(A_t = a) \right],$$

where we identify $\mathbb{E}[\mathbb{1}(A_t = a)] = \mathbb{E}[p_t^a]$. Since $p_1^a = 1/K$ by definition, we need to bound

$$\mathbb{E} \left[ \sum_{t=2}^{T} p_t^a \right] = \mathbb{E} \left[ \sum_{t=2}^{T} \mathbb{E}[p_t^a] \right].$$

Consider first the case where the learning rate is $\eta_t = \sqrt{\frac{\ln K}{\max_a S_{t-1}(a) + (K-1)^2}}$. We bound the individual probabilities as :

$$
\begin{aligned}
p_t^a &= \frac{\exp\left(-\eta_t D_{t-1}(a) - \eta_t^2 S_{t-1}(a)\right)}{\sum_{a=1}^{K} \exp\left(-\eta_t D_{t-1}(a) - \eta_t^2 S_{t-1}(a)\right)} \\
&= \frac{\exp\left(-\eta_t(D_{t-1}(a) - D_{t-1}(a^\star)) - \eta_t^2(S_{t-1}(a) - S_{t-1}(a^\star))\right)}{\sum_{a=1}^{K} \exp\left(-\eta_t(D_{t-1}(a) - D_{t-1}(a^\star)) - \eta_t^2(S_{t-1}(a) - S_{t-1}(a^\star))\right)} \\
&\leq \exp\left(-\eta_t(D_{t-1}(a) - D_{t-1}(a^\star)) - \eta_t^2(S_{t-1}(a) - S_{t-1}(a^\star))\right) \\
&\leq \exp\left(-\eta_t(D_{t-1}(a) - D_{t-1}(a^\star))\right) \exp\left(\eta_t^2 S_{t-1}(a^\star)\right) \\
&\leq K \exp\left(-\eta_t \sum_{i=1}^{t-1} X_i\right),
\end{aligned}
\tag{21}
$$

where we have defined $\sum X_i = D_{t-1}^a - D_{t-1}^{a^\star}$ and used $\eta_t^2 S_{t-1}(a^\star) \leq \ln K$.

Next we split up the expectation in two parts around a threshold $\sigma > 0$, using $p_t^a \leq 1$ and (21):

$$\mathbb{E}[p_t^a] \leq \sigma \, \mathbb{P}\{p_t^a \leq \sigma\} + 1 \cdot \mathbb{P}\{p_t^a > \sigma\} \leq \sigma + \mathbb{P}\left\{K \exp\left(-\eta_t \sum_{i=1}^{t-1} X_i\right) > \sigma\right\}, \tag{22}$$

Since $\eta_t$ is a random variable correlated with the $X_i$'s, we cannot directly bound this expression. We can however split the event under the probability into two separate cases, and upper bound the expression using upper and lower bounds on $\eta_t$ in the cases where $\sum X_i$ is negative or positive:

$$
\begin{aligned}
\mathbb{P}\left\{K \exp\left(-\eta_t \sum_{i=1}^{t-1} X_i\right) > \sigma\right\} &= \mathbb{P}\left\{K \exp\left(-\eta_t \sum_{i=1}^{t-1} X_i\right) > \sigma \,\&\, \sum_{i=1}^{t-1} X_i \leq 0\right\} \\
&\quad + \mathbb{P}\left\{K \exp\left(-\eta_t \sum_{i=1}^{t-1} X_i\right) > \sigma \,\&\, \sum_{i=1}^{t-1} X_i > 0\right\} \\
&\leq \mathbb{P}\left\{K \exp\left(-\bar{\eta}_t \sum_{i=1}^{t-1} X_i\right) > \sigma\right\} \\
&\quad + \mathbb{P}\left\{K \exp\left(-\frac{\sum_{i=1}^{t-1} X_i}{2(K-1)}\right) > \sigma\right\},
\end{aligned}
$$

where we have introduced $\bar{\eta}_t := \sqrt{\frac{\ln K}{(t-1)\varepsilon^2 + 1}} \frac{1}{K-1}$, which is a lower bound on $\eta_t$. Introducing $E = \sum_i \mathbb{E}_{B_i}[X_i] = (t-1)(K-1)\Delta_a$ and the shorthand $V = \sum_{i=1}^{t-1} X_i - E$, we can rewrite the probabilities, resulting in

$$\mathbb{E}[p_t^a] \leq \sigma + \mathbb{P}\left\{V < -\frac{\ln(\sigma/K)}{\bar{\eta}_t} - E\right\} + \mathbb{P}\left\{V < -2(K-1)\ln(\sigma/K) - E\right\}.$$

Since $V$ is the sum of martingale difference sequences we want to use Azuma's inequality, which requires that the right hand sides are negative. Choosing a positive splitting point $\sigma$ as

$$\sigma = K \exp\left(-\frac{(t-1)\Delta_a}{2}\sqrt{\frac{\ln K}{(t-1)\varepsilon^2 + 1}}\right), \tag{23}$$

the two right hand sides become

$$-\frac{\ln(\sigma/K)}{\bar{\eta}_t} - E = -\frac{E}{2}, \tag{24}$$

$$-2(K-1)\ln(\sigma/K) - E = E\left(\sqrt{\frac{\ln K}{(t-1)\varepsilon^2 + 1}} - 1\right) \le -\frac{E}{2}, \tag{25}$$

using $\sqrt{\frac{\ln K}{(t-1)\varepsilon^2 + 1}} = (K-1)\bar{\eta}_t \le 1/2$ for the final inequality. As these are negative, we can use Azuma's inequality which since the range of the $X_i$'s is $2(K-1)\varepsilon$ gives us

$$\mathbb{E}[p_t^a] \le K \exp\left(-\frac{(t-1)\Delta_a}{2}\sqrt{\frac{\ln K}{(t-1)\varepsilon^2 + 1}}\right) + 2\exp\left(-\frac{E^2/4}{2(t-1)(K-1)^2\varepsilon^2}\right), \tag{26}$$

where the inequality comes from substitution of (23), (24) and (25), and the two probabilities becomes one expression using the final inequality of (25).

We now consider two cases of the first term in (26). If $(t-1)\varepsilon^2 \ge 1$, then

$$\exp\left(-\frac{(t-1)\Delta_a}{2}\sqrt{\frac{\ln K}{(t-1)\varepsilon^2 + 1}}\right) \le \exp\left(-\frac{1}{2}\sqrt{\frac{\ln K}{2}}\frac{\Delta_a}{\varepsilon}\sqrt{t-1}\right).$$

If instead $(t-1)\varepsilon^2 \le 1$, then

$$\exp\left(-\frac{(t-1)\Delta_a}{2}\sqrt{\frac{\ln K}{(t-1)\varepsilon^2 + 1}}\right) \le \exp\left(-\frac{\Delta_a}{2}\sqrt{\frac{\ln K}{2}}t\right).$$

For both cases the second term in (26) becomes

$$2\exp\left(-\frac{1}{8}\frac{\Delta_a^2}{\varepsilon^2}(t-1)\right).$$

For $\eta_t = \frac{1}{2(K-1)}$, we first note that $\eta_t \le \sqrt{\frac{\ln K}{\max_a S_{t-1}(a) + (K-1)^2}}$, so the bound used for $p_t^a$ in (22) still applies. Since $\eta_t$ is no longer a random variable, we have

$$\mathbb{E}[p_t^a] \le \sigma + \mathbb{P}\left\{K\exp\left(-\frac{\sum X_i}{2(K-1)}\right) > \sigma\right\}.$$

Rewriting this as before and choosing $\sigma = K\exp\left(-\frac{(t-1)\Delta_a}{4}\right)$, we get by Azuma's inequality

$$\mathbb{E}[p_t^a] \le K\exp\left(-\frac{(t-1)\Delta_a}{4}\right) + \exp\left(-\frac{1}{8}\frac{\Delta_a^2}{\varepsilon^2}(t-1)\right).$$

We now have three cases of bounds on $\mathbb{E}[p_t^a]$. For each of these the analysis is completed by summing over $t = 2$ to $\infty$, using Lemma 3 and then summing over the arms times the gaps. For all cases, the result is smaller than the right hand side in Theorem 3. $\qquad\square$