[Reviews · NeurIPS 2018]

Reviewer 1



Summary This paper considers the prediction with limited advice game, in which a forecaster is to predict an action, but only sees the loss for that action and and the losses for a limited number of other actions. The algorithm that is developed, SODA, is able to adapt to two types of easiness: stochastic loss sequences and adversarial loss sequences with small effective loss range. One contribution of this paper is to show that with only one extra observation compared to the bandit setting SODA is able to bypass the impossibility of being adaptive to the effective loss range in the bandit setting. This extra observation is used to estimate the loss differences between the actions and used in a similar manner as Wintenberger (2017) uses the true loss differences. This paper also provides lower bounds in the adversarial setting to show that the regret of SODA matches the lower bound up to logarithmic factors and lower order factors. Furthermore, SODA achieves constant regret for stochastic loss sequences, with suboptimal scaling in the total number of actions. Strengths and weaknesses The paper was pleasant to read and clearly written. SODA is based on standard exponential weights with second order corrections, but with SODA the loss differences are estimated instead of observed. The estimates of the loss differences are based on an extra randomly chosen observation, which seems to be a novel approach. However, the extra observation is based on uniform draws from the remaining actions after choosing which action is played in the current round. As the authors note it seems unlikely that the regret guarantees for adversarial and stochastic sequences can not be improved by sampling the additional observation in a more clever manner. Especially for stochastic losses it appears there is room for improvement. Also, it would be interesting to see how allowing for more than one extra observation improves the regret. Nevertheless, with limited feedback being able to adapt to the two types of easiness simultaneously is a significant contribution. Minor Comments - line 84: suffered.) -> suffered). - in line 210 of the "full version including appendices" version of the paper the equation numbers are replaced by ??. In the rebuttal, the authors cleared up any remaining confusion.

Reviewer 2



The paper presents an novel online algorithm for stochastic losses and adversarial losses with a small range under the assumption that they can observe an additional loss per round. In the adversarial setting, they provide a lower bound that almost matched their upper bound. In the stochastic setting, their regret bound remarkably does not depend on the rounds T. Even though the setting described in the paper can be characterized by a sequence of feedback graphs, the authors do not cite or compare to the large body of literature of online learning with feedback graphs (i.e. "Online Learning with Feedback Graphs: Beyond Bandits" by Alon at al. 2015, "Online Learning with Feedback Graphs Without the Graphs" Cohen et al. 2016, "Leveraging Side Observations in Stochastic Bandits" by Caron et al. 2012). Leveraging Side Observations in Stochastic Bandits by Caron et al. 2012 prove a lower bound of log(T) for stochastic losses when the feedback graph is fixed. The feedback graph used in the current paper is more complex than a fixed graph so it is surprising that they can attain a bound that does not depend on log T. I am not sure how the lower bound of Caron et al. can be reconciled with the stochastic bound in the paper, but it might be because in the current paper, they are allowed to chose which second arm they want to observe. Nevertheless, a thorough discussion and analysis is warranted here. Another criticism is that they claim their assumptions are weak. Via their assumptions, they can bypass the impossibility result of Gerchinovitz and Lattimore [2016], but allowing an extra loss observation is essentially changing the setting and moreover they allow the algorithm to chose the second arm that they get to observe. Adding extra loss observations even if it's just one more at each round can be hugely beneficial as shown in the above feedback graphs papers. The paper is understandable, but not well written or polished. Due to the lack of comparison to on-line learning with feedback graphs and the fact that Caron et al. 2012 have a lower bound that depends on log(T), I cannot vote to accept unless these issues are addressed by the authors in the rebuttal. ----------------------------------------------------------------------------------------------- The authors have adequately addressed my concerns and hence I increased their overall score.

Reviewer 3



This paper studies the problem of achieving a regret scaled with the effective range of the losses in the bandit setting. Although it has been shown that it is impossible in the standard bandit setting, this paper proposes the second order difference adjustments algorithm (SODA) which achieves the goal when one random extra observation is available for each round in the bandit setting. Such result improves the previous one by Cesa-Bianchi and Shamir which requires the availability of the smallest loss for each round. More desirable, SODA is also proved to achieve constant expected regret in the stochastic setting, thus it automatically adapts between two regimes. Other comments: 1. On line 210, equation numbers are missing. 2. In the proof of Theorem 1, line 192. The analysis about the 3rd term is missing: the expectation of this term is 0. 3. The ‘a’ In Equation (10) and ‘a’ in Equation (11) collide. Same on line 329. 4. There is still a gap between dependencies on K in the lower bound and in the upper bound. Any comment on it?